# Local Similarity-Aware Deep Feature Embedding

**Chen Huang    Chen Change Loy    Xiaoou Tang**
Department of Information Engineering, The Chinese University of Hong Kong
{chuang,ccloy,xtang}@ie.cuhk.edu.hk

## Abstract

Existing deep embedding methods in vision tasks are capable of learning a compact Euclidean space from images, where Euclidean distances correspond to a similarity metric. To make learning more effective and efficient, hard sample mining is usually employed, with samples identified through computing the Euclidean feature distance. However, the global Euclidean distance cannot faithfully characterize the true feature similarity in a complex visual feature space, where the intraclass distance in a high-density region may be larger than the interclass distance in low-density regions. In this paper, we introduce a Position-Dependent Deep Metric (PDDM) unit, which is capable of learning a similarity metric adaptive to local feature structure. The metric can be used to select genuinely hard samples in a local neighborhood to guide the deep embedding learning in an online and robust manner. The new layer is appealing in that it is pluggable to any convolutional networks and is trained end-to-end. Our local similarity-aware feature embedding not only demonstrates faster convergence and boosted performance on two complex image retrieval datasets, its large margin nature also leads to superior generalization results under the large and open set scenarios of transfer learning and zero-shot learning on ImageNet 2010 and ImageNet-10K datasets.

## 1 Introduction

Deep embedding methods aim at learning a compact feature embedding $f(x) \in \mathbb{R}^d$ from image $x$ using a deep convolutional neural network (CNN). They have been increasingly adopted in a variety of vision tasks such as product visual search [1, 14, 29, 33] and face verification [13, 27]. The embedding objective is usually in a Euclidean sense: the Euclidean distance $D_{i,j} = \|f(x_i) - f(x_j)\|_2$ between two feature vectors should preserve their semantic relationship encoded pairwise (by contrastive loss [1]), in triplets [27, 33] or even higher order relationships (*e.g.*, by lifted structured loss [29]).

It is widely observed that an effective data sampling strategy is crucial to ensure the quality and learning efficiency of deep embedding, as there are often many more easy examples than those meaningful hard examples. Selecting overly easy samples can in practice lead to slow convergence and poor performance since many of them satisfy the constraint well and give nearly zero loss, without exerting any effect on parameter update during the back-propagation [3]. Hence hard example mining [7] becomes an indispensable step in state-of-the-art deep embedding methods. These methods usually choose hard samples by computing the convenient Euclidean distance in the embedding space. For instance, in [27, 29], hard negatives with small Euclidean distances are found online in a mini-batch. An exception is [33] where an online reservoir importance sampling scheme is proposed to sample discriminative triplets by relevance scores. Nevertheless, these scores are computed offline with different hand-crafted features and distance metrics, which is suboptimal.

We question the effectiveness of using a *single* and *global* Euclidean distance metric for finding hard samples, especially for real-world vision tasks that exhibit complex feature variations due to pose, lighting, and appearance. As shown in a fine-grained bird image retrieval example in Figure 1(a), the diversity of feature patterns learned for each class throughout the feature space can easily lead

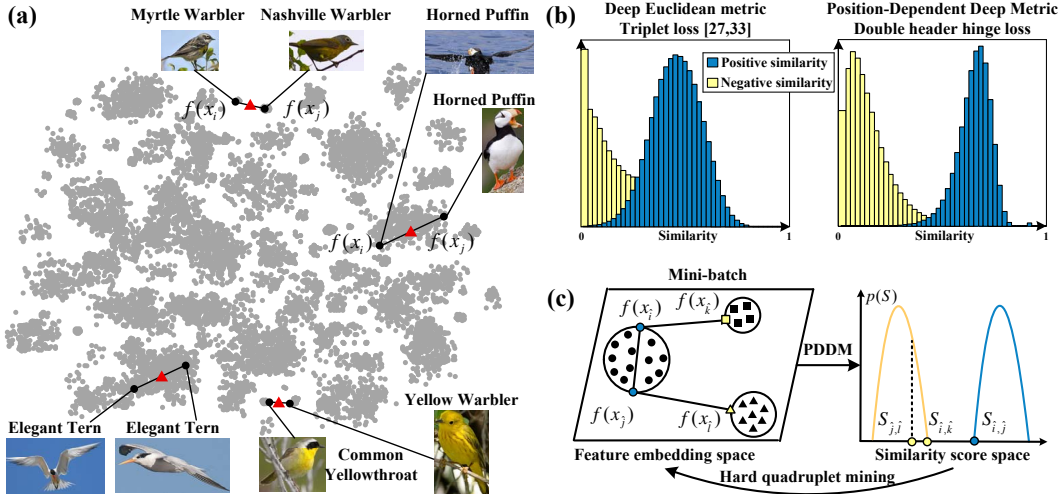

Figure 1: (a) 2-D feature embedding (by t-SNE [19]) of the CUB-200-2011 [32] test set. The intraclass distance can be larger than the interclass distance under the global Euclidean metric, which can mislead the hard sample mining and consequently deep embedding learning. We propose a PDDM unit that incorporates the absolute position (*i.e.*, feature mean denoted by the red triangle) to adapt metric to the local feature structure. (b) Overlapped similarity distribution by the Euclidean metric (the similarity scores are transformed from distances by a Sigmoid-like function) vs. the well-separated distribution by PDDM. (c) PDDM-guided hard sample mining and embedding learning.

to a larger intraclass Euclidean distance than the interclass distance. Such a heterogeneous feature distribution yields a highly overlapped similarity score distribution for the positive and negative pairs, as shown in the left chart of Figure 1(b). We observed similar phenomenon for the global Mahalanobis metric [11, 12, 21, 35, 36] in our experiments. It is not difficult to see that using a single and global metric would easily mislead the hard sample mining. To circumvent this issue, Cui *et al.* [3] resorted to human intervention for harvesting genuinely hard samples.

Mitigating the aforementioned issue demands an improved metric that is adaptive to the local feature structure. In this study we wish to learn a local-adaptive similarity metric online, which will be exploited to search for high-quality hard samples in local neighborhood to facilitate a more effective deep embedding learning. Our key challenges lie in the formulation of a new layer and loss function that jointly consider the similarity metric learning, hard samples selection, and deep embedding learning. Existing studies [13, 14, 27, 29, 33] only consider the two latter objectives but not together with the first. To this end, we propose a new Position-Dependent Deep Metric (PDDM) unit for similarity metric learning. It is readily pluggable to train end-to-end with an existing deep embedding learning CNN. We formulate the PDDM such that it learns locally adaptive metric (unlike the global Euclidean metric), through a non-linear regression on both the absolute feature difference and feature mean (which encodes absolute position) of a data pair. As depicted in the right chart of Figure 1(b), the proposed metric yields a similarity score distribution that is more distinguishable than the conventional Euclidean metric. As shown in Figure 1(c), hard samples are mined from the resulting similarity score space and used to optimize the feature embedding space in a seamless manner. The similarity metric learning in PDDM and embedding learning in the associated CNN are jointly optimized using a novel large-margin *double-header hinge loss*.

Image retrieval experiments on two challenging real-world vision datasets, CUB-200-2011 [32] and CARS196 [15], show that our local similarity-aware feature embedding significantly outperforms state-of-the-art deep embedding methods that come without the online metric learning and associated hard sample mining scheme. Moreover, the proposed approach incurs a far lower computational cost and encourages faster convergence than those structured embedding methods (*e.g.*, [29]), which need to compute a fully connected dense matrix of pairwise distances in a mini-batch. We further demonstrate our learned embedding is generalizable to new classes in large open set scenarios. This is validated in the transfer learning and zero-shot learning (using the ImageNet hierarchy as auxiliary knowledge) tasks on ImageNet 2010 and ImageNet-10K [5] datasets.

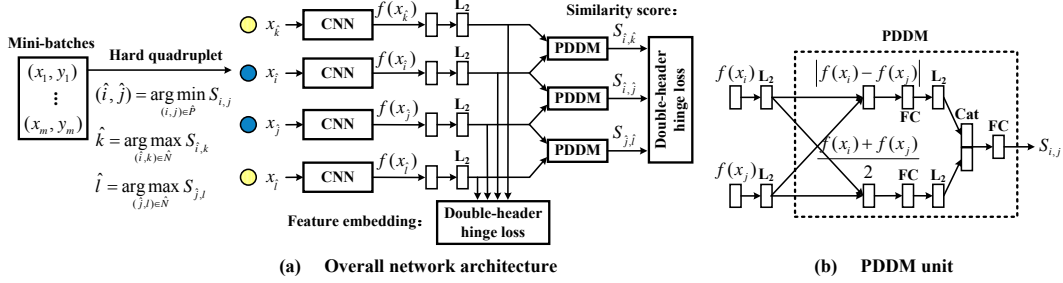

Figure 2: (a) The overall network architecture. All CNNs have shared architectures and parameters. (b) The PDDM unit.

## 2 Related work

**Hard sample mining in deep learning:** Hard sample mining is a popular technique used in computer vision for training robust classifier. The method aims at augmenting a training set progressively with false positive examples with the model learned so far. It is the core of many successful vision solutions, *e.g.* pedestrian detection [4, 7]. In a similar spirit, contemporary deep embedding methods [27, 29] choose hard samples in a mini-batch by computing the Euclidean distance in the embedding space. For instance, Schroff *et al.* [27] selected online the semi-hard negative samples with relatively small Euclidean distances. Wang *et al.* [33] proposed an online reservoir importance sampling algorithm to sample triplets by relevance scores, which are computed offline with different distance metrics. Similar studies on image descriptor learning [28] and unsupervised feature learning [34] also select hard samples according to the Euclidean distance-based losses in their respective CNNs. We argue in this paper that the global Euclidean distance is a suboptimal similarity metric for hard sample mining, and propose a locally adaptive metric for better mining.

**Metric learning:** An effective similarity metric is at the core of hard sample mining. Euclidean distance is the simplest similarity metric, and it is widely used by current deep embedding methods where Euclidean feature distances directly correspond the similarity. Similarities can be encoded pairwise with a contrastive loss [1] or in more flexible triplets [27, 33]. Song *et al.* [29] extended to even higher order similarity constraints by lifting the pairwise distances within a mini-batch to the dense matrix of pairwise distances. Beyond Euclidean metric, one can actually turn to the parametric Mahalanobis metric instead. Representative works [12, 36] minimize the Mahalanobis distance between positive sample pairs while maximizing the distance between negative pairs. Alternatives directly optimize the Mahalanobis metric for nearest neighbor classification via the method of Neighbourhood Component Analysis (NCA) [11], Large Margin Nearest Neighbor (LMNN) [35] or Nearest Class Mean (NCM) [21]. However, the common drawback of the Mahalanobis and Euclidean metrics is that they are both global and are far from being ideal in the presence of heterogeneous feature distribution (see Figure 1(a)). An intuitive remedy would be to learn multiple metrics [9], which would be computationally expensive though. Xiong *et al.* [37] proposed a single adaptive metric using the absolute position information in random forest classifiers. Our approach shares the similar intuition, but incorporates the position information by a deep CNN in a more principled way, and can jointly learn similarity-aware deep features instead of using hand-crafted ones as in [37].

## 3 Local similarity-aware deep embedding

Let $X = \{(x_i, y_i)\}$ be an imagery dataset, where $y_i$ is the class label of image $x_i$. Our goal is to jointly learn a deep feature embedding $f(x)$ from image $x$ into a feature space $\mathbb{R}^d$, and a similarity metric $S_{i,j} = S(f(x_i), f(x_j)) \in \mathbb{R}^1$, such that the metric can robustly select hard samples online to learn a discriminative, similarity-aware feature embedding. Ideally, the learned features $(f(x_i), f(x_j))$ from the set of positive pairs $P = \{(i,j)|y_i = y_j\}$ should be close to each other with a large similarity score $S_{i,j}$, while the learned features from the set of negative pairs $N = \{(i,j)|y_i \neq y_j\}$ should be pushed far away with a small similarity score. Importantly, this relationship should hold independent of the (heterogeneous) feature distribution in $\mathbb{R}^d$, where a global metric $S_{i,j}$ can fail. To adapt $S_{i,j}$ to the latent structure of feature embeddings, we propose a Position-Dependent Deep Metric (PDDM) unit that can be trained end-to-end, see Figure 2(b).

The overall network architecture is shown in Figure 2(a). First, we use PDDM to compute similarity scores for the mini-batch samples during a particular forward pass. The scores are used to select one hard quadruplet from the local sets of positive pairs $\hat{P} \in P$ and negative pairs $\hat{N} \in N$ in the batch. Then each sample in the hard quadruplet is separately fed into four identical CNNs with shared parameters $W$ to extract $d$-dimensional features. Finally, a discriminative double-header hinge loss is applied to both the similarity score and feature embeddings. This enables us to jointly optimize the two that benefit each other. We will provide the details in the following.

## 3.1 PDDM learning and hard sample mining

Given a feature pair $(f_W(x_i), f_W(x_j))$ extracted from images $x_i$ and $x_j$ by an embedding function $f_W(\cdot)$ parameterized by $W$, we wish to obtain an ideal similarity score $y_{i,j} = 1$ if $(i,j) \in P$, and $y_{i,j} = 0$ if $(i,j) \in N$. Hence, we seek the optimal similarity metric $S^*(\cdot, \cdot)$ from an appropriate function space $H$, and also seek the optimal feature embedding parameters $W^*$:

$$(S^*(\cdot, \cdot), W^*) = \underset{S(\cdot, \cdot) \in H, W}{\operatorname{argmin}} \frac{1}{|P \cup N|} \sum_{(i,j) \in P \cup N} l\left(S(f_W(x_i), f_W(x_j)), y_{i,j}\right), \qquad (1)$$

where $l(\cdot)$ is some loss function. We will omit the parameters $W$ of $f(\cdot)$ in the following for brevity.

**Adapting to local feature structure**. The standard Euclidean or Mahalanobis metric can be seen as a special form of function $S(\cdot, \cdot)$ that is based solely on the feature difference vector $u = |f(x_i) - f(x_j)|$ or its linearly transformed version. These metrics are suboptimal in a heterogeneous embedding space, thus could easily fail the searching of genuinely hard samples. On the contrary, the proposed PDDM leverages the absolute feature position to adapt the metric throughout the embedding space. Specifically, inspired by [37], apart from the feature difference vector $u$, we additionally incorporate the feature mean vector $v = (f(x_i) + f(x_j))/2$ to encode the absolute position. Unlike [37], we formulate a principled *learnable* similarity metric from $u$ and $v$ in our CNN.

Formally, as shown in Figure 2(b), we first normalize the features $f(x_i)$ and $f(x_j)$ onto the unit hypersphere, *i.e.*, $\|f(x)\|_2 = 1$, in order to maintain feature comparability. Such normalized features are used to compute their relative and absolute positions encoded in $u$ and $v$, each followed by a fully connected layer, an elementwise ReLU nonlinear function $\sigma(\xi) = \max(0, \xi)$, and again, $\ell_2$-normalization $r(x) = x/\|x\|_2$. To treat $u$ and $v$ differently, the fully connected layers applied to them are not shared, parameterized by $W_u \in \mathbb{R}^{d \times d}, b_u \in \mathbb{R}^d$ and $W_v \in \mathbb{R}^{d \times d}, b_v \in \mathbb{R}^d$, respectively. The nonlinearities ensure the model is not trivially equivalent to be the mapping from $f(x_i)$ and $f(x_j)$ themselves. Then we concatenate the mapped $u'$ and $v'$ vectors and pass them through another fully connected layer parameterized by $W_c \in \mathbb{R}^{2d \times d}, b_c \in \mathbb{R}^d$ and the ReLU function, and finally map to a score $S_{i,j} = S(f(x_i), f(x_j)) \in \mathbb{R}^1$ via $W_s \in \mathbb{R}^{d \times 1}, b_s \in \mathbb{R}^1$. To summarize:

$$u = |f(x_i) - f(x_j)|, \quad v = (f(x_i) + f(x_j))/2,$$
$$u' = r\left(\sigma(W_u u + b_u)\right), \quad v' = r\left(\sigma(W_v v + b_v)\right),$$
$$c = \sigma\left(W_c \begin{bmatrix} u' \\ v' \end{bmatrix} + b_c\right), \quad S_{i,j} = W_s c + b_s. \qquad (2)$$

In this way, we transform the seeking of the similarity metric function $S(\cdot, \cdot)$ into the joint learning of CNN parameters ($W_u, W_v, W_c, W_s, b_u, b_v, b_c, b_s$ for the PDDM unit, and $W$ for feature embeddings). The parameters collectively define a flexible nonlinear regression function for the similarity score.

**Double-header hinge loss**. To optimize all these CNN parameters, we can choose a standard regression loss function $l(\cdot)$, *e.g.*, logistic regression loss. Or alternatively, we can cast the problem as a binary classification one as in [37]. However, in both cases the CNN is prone to overfitting, because the supervisory binary similarity labels $y_{i,j} \in \{0, 1\}$ tend to independently push the scores towards two single points. While in practice, the similarity scores of positive and negative pairs live on a 1-D manifold following some distribution patterns on heterogeneous data, as illustrated in Figure 1(b).

This motivates us to design a loss function $l(\cdot)$ to separate the similarity *distributions*, instead of in an independent pointwise way that is noise-sensitive. One intuitive option is to impose the Fisher criterion [20] on the similarity scores, *i.e.*, maximizing the ratio between the interclass and intraclass scatters of scores. Similarly, it can be reduced to maximize $(\mu_P - \mu_N)^2/(Var_P + Var_N)$ in our 1-D case, where $\mu$ and *Var* are the mean and variance of each score distribution. Unfortunately, the

optimality of Fisher-like criteria relies on the assumption that the data of each class is of a Gaussian distribution, which is obviously not satisfied in our case. Also, a high cost $O(m^2)$ is entailed to compute the Fisher loss in a mini-batch with $m$ samples by computing all the pairwise distances.

Consequently, we propose a faster-to-compute loss function that approximately maximizes the margin between the positive and negative similarity distributions without making any assumption about the distribution's shape or pattern. Specifically, we retrieve *one hard quadruplet* from a random batch during each forward pass. Please see the illustration in Figure 1(c). The quadruplet consists of the most dissimilar positive sample pair in the batch $(\hat{i}, \hat{j}) = \mathrm{argmin}_{(i,j)\in\hat{P}} S_{i,j}$, which means their similarity score is most likely to cross the "safe margin" towards the negative similarity distribution in this local range. Next, we build a similarity neighborhood graph that links the chosen positive pair with their respective negative neighbors in the batch, and choose the hard negatives as the other two quadruplet members $\hat{k} = \mathrm{argmax}_{(\hat{i},k)\in\hat{N}} S_{\hat{i},k}$, and $\hat{l} = \mathrm{argmax}_{(\hat{j},l)\in\hat{N}} S_{\hat{j},l}$. Using this hard quadruplet $(\hat{i}, \hat{j}, \hat{k}, \hat{l})$, we can now locally approximate the inter-distribution margin as $\min(S_{\hat{i},\hat{j}} - S_{\hat{i},\hat{k}}, S_{\hat{i},\hat{j}} - S_{\hat{j},\hat{l}})$ in a robust manner. This makes us immediately come to a double-header hinge loss $E_m$ to discriminate the target similarity distributions under the large margin criterion:

$$\min \quad E_m = \sum\nolimits_{\hat{i},\hat{j}} (\varepsilon_{\hat{i},\hat{j}} + \tau_{\hat{i},\hat{j}}), \tag{3}$$

$$s.t. : \forall(\hat{i},\hat{j}), \quad \max\left(0, \alpha + S_{\hat{i},\hat{k}} - S_{\hat{i},\hat{j}}\right) \le \varepsilon_{\hat{i},\hat{j}}, \ \max\left(0, \alpha + S_{\hat{j},\hat{l}} - S_{\hat{i},\hat{j}}\right) \le \tau_{\hat{i},\hat{j}},$$

$$(\hat{i},\hat{j}) = \mathop{\mathrm{argmin}}_{(i,j)\in\hat{P}} S_{i,j}, \ \hat{k} = \mathop{\mathrm{argmax}}_{(\hat{i},k)\in\hat{N}} S_{\hat{i},k}, \ \hat{l} = \mathop{\mathrm{argmax}}_{(\hat{j},l)\in\hat{N}} S_{\hat{j},l}, \ \varepsilon_{\hat{i},\hat{j}} \ge 0, \ \tau_{\hat{i},\hat{j}} \ge 0,$$

where $\varepsilon_{\hat{i},\hat{j}}, \tau_{\hat{i},\hat{j}}$ are the slack variables, and $\alpha$ is the enforced margin.

The discriminative loss has four main benefits: 1) The discrimination of similarity distributions is assumption-free. 2) Hard samples are simultaneously found during the loss minimization. 3) The loss function incurs a low computational cost and encourages faster convergence. Specifically, the searching cost of the hard positive pair $(\hat{i}, \hat{j})$ is very small since the positive pair set $\hat{P}$ is usually much smaller than the negative pair set $\hat{N}$ in an $m$-sized mini-batch. While the hard negative mining only incurs an $O(m)$ complexity. 4) Eqs. (2, 3) can be easily optimized through the standard stochastic gradient descent to adjust the CNN parameters.

## 3.2 Joint metric and embedding optimization

Given the learned PDDM and mined hard samples in a mini-batch, we can use them to solve for a better, local similarity-aware feature embedding at the same time. For computational efficiency, we reuse the hard quadruplet's features for metric optimization (Eq. (3)) in the same forward pass. What follows is to use the double-header hinge loss again, but to constrain the deep features this time, see Figure 2. The objective is to ensure the Euclidean distance between hard negative features ($D_{\hat{i},\hat{k}}$ or $D_{\hat{j},\hat{l}}$) is larger than that between hard positive features $D_{\hat{i},\hat{j}}$ by a large margin. Combining the embedding loss $E_e$ and metric loss $E_m$ (Eq. (3)) gives our final joint loss function:

$$\min \quad E_m + \lambda E_e + \gamma \|\widetilde{W}\|_2, \ \text{where } E_e = \sum\nolimits_{\hat{i},\hat{j}} (o_{\hat{i},\hat{j}} + \rho_{\hat{i},\hat{j}}), \tag{4}$$

$$s.t. : \forall(\hat{i},\hat{j}), \quad \max\left(0, \beta + D_{\hat{i},\hat{j}} - D_{\hat{i},\hat{k}}\right) \le o_{\hat{i},\hat{j}}, \ \max\left(0, \beta + D_{\hat{i},\hat{j}} - D_{\hat{j},\hat{l}}\right) \le \rho_{\hat{i},\hat{j}},$$

$$D_{\hat{i},\hat{j}} = \|f(x_{\hat{i}}) - f(x_{\hat{j}})\|_2, \ o_{\hat{i},\hat{j}} \ge 0, \ \rho_{\hat{i},\hat{j}} \ge 0,$$

where $\widetilde{W}$ are the CNN parameters for both the PDDM and feature embedding, and $o_{\hat{i},\hat{j}}, \rho_{\hat{i},\hat{j}}$ and $\beta$ are the slack variables and enforced margin for $E_e$, and $\lambda, \gamma$ are the regularization parameters. Since all features are $\ell_2$-normalized (see Figure 2), we have $\beta + D_{\hat{i},\hat{j}} - D_{\hat{i},\hat{k}} = \beta - 2f(x_{\hat{i}})f(x_{\hat{j}}) + 2f(x_{\hat{i}})f(x_{\hat{k}})$, and can conveniently derive the gradients as those in triplet-based methods [27, 33].

This joint objective provides effective supervision in two domains, respectively at the score level and feature level that are mutually informed. Although the score level supervision by $E_m$ alone is already capable of optimizing both our metric and feature embedding, we will show the benefits of adding the feature level supervision by $E_e$ in experiments. Note we can still enforce the large margin relations of quadruple features in $E_e$ using the simple Euclidean metric. This is because the quadruple features are selected by our PDDM that is learned in the local Euclidean space as well.

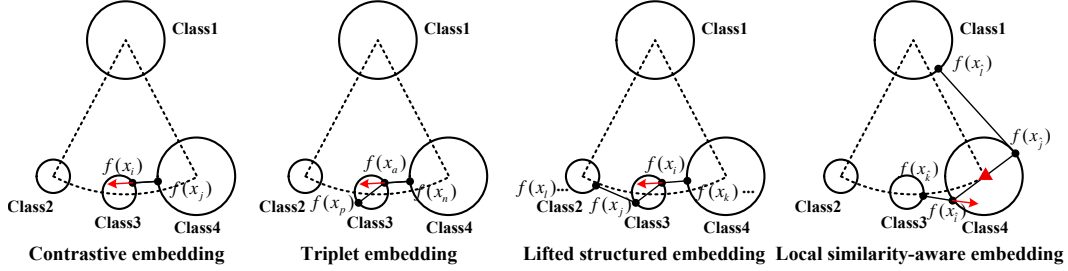

Figure 3: Illustrative comparison of different feature embeddings. Pairwise similarities in classes 1-3 are effortlessly distinguishable in a heterogeneous feature space because there is always a *relative* safe margin between any two involved classes w.r.t. their class bounds. However, it is not the case for class 4. The contrastive [1], triplet [27, 33] and lifted structured [29] embeddings select hard samples by the Euclidean distance that is not adaptive to the local feature structure. They may thus select inappropriate hard samples and the negative pairs get misled towards the wrong gradient direction (red arrow). In contrast, our local similarity-aware embedding is correctly updated by the genuinely hard examples in class 4.

Figure 3 compares our local similarity-aware feature embedding with existing works. Contrastive [1] embedding is trained on pairwise data $\{(x_i, x_j, y_{i,j})\}$, and tries to minimize the distance between the positive feature pair and penalize the distance between negative feature pair for being smaller than a margin $\alpha$. Triplet embedding [27, 33] samples the triplet data $\{(x_a, x_p, x_n)\}$ where $x_a$ is an anchor point and $x_p, x_n$ are from the same and different class, respectively. The objective is to separate the intraclass distance between $(f(x_a), f(x_p))$ and interclass distance between $(f(x_a), f(x_n))$ by margin $\alpha$. While lifted structured embedding [29] considers all the positive feature pairs (*e.g.*, $(f(x_i), f(x_j))$ in Figure 3) and all their linked negative pairs (*e.g.*, $(f(x_i), f(x_k)), (f(x_j), f(x_l))$ and so on), and enforces a margin $\alpha$ between positive and negative distances.

The common drawback of the above-mentioned embedding methods is that they sample pairwise or triplet (*i.e.*, anchor) data randomly and rely on simplistic Euclidean metric. They are thus very likely to update from inappropriate hard samples and push the negative pairs towards the already well-separated embedding space (see the red arrow in Figure 3). While our method can use PDDM to find the genuinely hard feature quadruplet $(f(x_{\hat{i}}), f(x_{\hat{j}}), f(x_{\hat{k}}), f(x_{\hat{l}}))$, thus can update feature embedding in the correct direction. Also, our method is more efficient than the lifted structured embedding [29] that requires computing dense pairwise distances within a mini-batch.

### 3.3 Implementation details

We use GoogLeNet [31] (feature dimension $d = 128$) and CaffeNet [16] ($d = 4096$) as our base network architectures for retrieval and transfer learning tasks respectively. They are initialized with their pretrained parameters on ImageNet classification. The fully-connected layers of our PDDM unit are initialized with random weights and followed by dropout [30] with $p = 0.5$. For all experiments, we choose by grid search the mini-batch size $m = 64$, initial learning rate $1 \times 10^{-4}$, momentum 0.9, margin parameters $\alpha = 0.5, \beta = 1$ in Eqs. (3, 4), and regularization parameters $\lambda = 0.5, \gamma = 5 \times 10^{-4}$ ($\lambda$ balances the metric loss $E_m$ against the embedding loss $E_e$). To find meaningful hard positives in our hard quadruplets, we ensure that any one class in a mini-batch has at least 4 samples. And, we always scale $S_{\hat{i}, \hat{j}}$ into the range $[0, 1]$ by the similarity graph in the batch. The entire network is trained for a maximum of 400 epochs until convergence.

## 4 Results

**Image retrieval**. The task of image retrieval is a perfect testbed for our method, where both the learned PDDM and feature embedding (under the Euclidean feature distance) can be used to find similar images for a query. Ideally, a good similarity metric should be query-adapted (*i.e.*, position-dependent), and both the metric and features should be able to generalize. We test these properties of our method on two popular fine-grained datasets with complex feature distribution. We deliberately make the evaluation more challenging by preparing training and testing sets that are disjoint in terms of class labels. Specifically, we use the CUB-200-2011 [32] dataset with 200 bird classes and 11,788

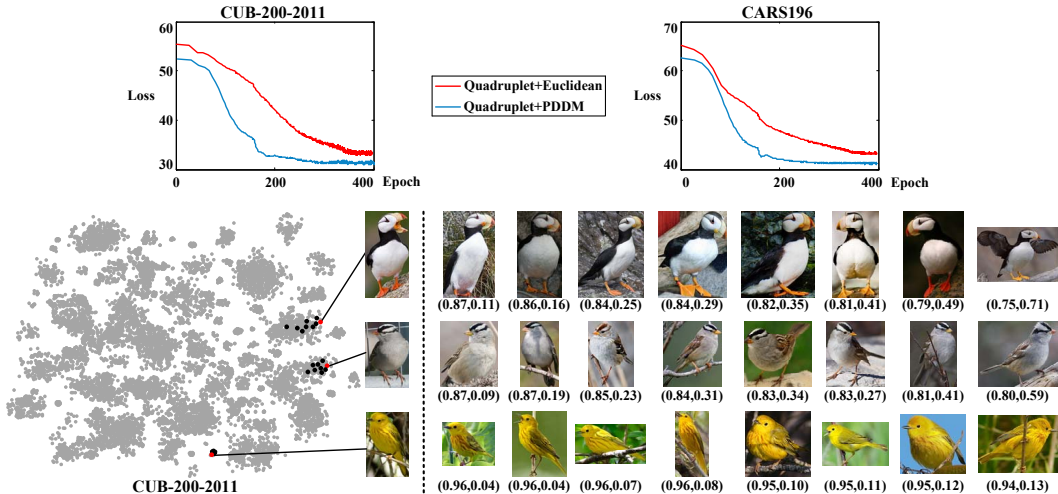

Figure 4: *Top*: a comparison of the training convergence curves of our method with Euclidean- and PDDM-based hard quadruplet mining on the test sets of CUB-200-2011 [32] and CARS196 [15] datasets. *Bottom*: top 8 images retrieved by PDDM (similarity score and feature distance are shown underneath) and the corresponding feature embeddings (black dots) on CUB-200-2011.

Table 1: Recall@K (%) on the test sets of CUB-200-2011 [32] and CARS196 [15] datasets.

| | CUB-200-2011 | | | | | | CARS196 | | | | | |
|---|---|---|---|---|---|---|---|---|---|---|---|---|
| K | 1 | 2 | 4 | 8 | 16 | 32 | 1 | 2 | 4 | 8 | 16 | 32 |
| Contrastive [1] | 26.4 | 37.7 | 49.8 | 62.3 | 76.4 | 85.3 | 21.7 | 32.3 | 46.1 | 58.9 | 72.2 | 83.4 |
| Triplet [27, 33] | 36.1 | 48.6 | 59.3 | 70.0 | 80.2 | 88.4 | 39.1 | 50.4 | 63.3 | 74.5 | 84.1 | 89.8 |
| LiftedStruct [29] | 47.2 | 58.9 | 70.2 | 80.2 | 89.3 | 93.2 | 49.0 | 60.3 | 72.1 | 81.5 | 89.2 | 92.8 |
| LMDM score | 49.5 | 61.1 | 72.1 | 81.8 | 90.5 | 94.1 | 50.9 | 61.9 | 73.5 | 82.5 | 89.8 | 93.1 |
| PDDM score | 55.0 | 67.1 | 77.4 | 86.9 | 92.2 | 95.0 | 55.2 | 66.5 | 78.0 | 88.2 | 91.5 | 94.3 |
| PDDM+Triplet | 50.9 | 62.1 | 73.2 | 82.5 | 91.1 | 94.4 | 46.4 | 58.2 | 70.3 | 80.1 | 88.6 | 92.6 |
| PDDM+Quadruplet | **58.3** | **69.2** | **79.0** | **88.4** | **93.1** | **95.7** | **57.4** | **68.6** | **80.1** | **89.4** | **92.3** | **94.9** |

images. We employ the first 100 classes (5,864 images) for training, and the remaining 100 classes (5,924 images) for testing. Another used dataset is CARS196 [15] with 196 car classes and 16,185 images. The first 98 classes (8,054 images) are used for training, and the other 98 classes are retained for testing (8,131 images). We use the standard Recall@K as the retrieval evaluation metric.

Figure 4-(top) shows that the proposed PDDM leads to $2\times$ faster convergence in 200 epochs (28 hours on a Titan X GPU) and lower converged loss than the regular Euclidean metric, when both are used to mine hard quadruplets for embedding learning. Note the two resulting approaches both incur lower computational costs than [29], with a near linear rather than quadratic [29] complexity in mini-batches. As observed from the retrieval results and their feature distributions in Figure 4-(bottom), our PDDM copes comfortably with large intraclass variations, and generates stable similarity scores for those differently scattered features positioned around a particular query. These results also demonstrate the successful generalization of PDDM on a test set with disjoint class labels.

Table 1 quantifies the advantages of both of our similarity metric (PDDM) and similarity-aware feature embedding (dubbed 'PDDM+Quadruplet' for short). In the middle rows, we compare the results from using the metrics of Large Margin Deep Metric (LMDM) and our PDDM, both jointly trained with our quadruplet embedding. The LMDM is implemented by deeply regressing the similarity score from the feature difference only, without using the absolute feature position. Although it is also optimized under the large margin rule, it performs worse than our PDDM due to the lack of position information for metric adaptation. In the bottom rows, we test using the learned features under the Euclidean distance. We observed PDDM significantly improves the performance of both triplet and quadruplet embeddings. In particular, our full 'PDDM+Quadruplet' method yields large gains (8%+ Recall@K=1) over previous works [1, 27, 29, 33] all using the Euclidean distance for hard sample mining. Indeed, as visualized in Figure 4, our learned features are typically well-clustered, with sharp boundaries and large margins between many classes.

Table 2: The flat top-1 accuracy (%) of transfer learning on ImageNet-10K [5] and flat top-5 accuracy (%) of zero-shot learning on ImageNet 2010.

| Transfer learning on ImageNet-10K | | | | | | Zero-shot learning on ImageNet 2010 | | | | | | |
|---|---|---|---|---|---|---|---|---|---|---|---|---|
| [5] | [26] | [23] | [18] | [21] | **Ours** | ConSE [22] | DeViSE [8] | PST [24] | [25] | [21] | AMP [10] | **Ours** |
| 6.4 | 16.7 | 18.1 | 19.2 | 21.9 | **28.4** | 28.5 | 31.8 | 34.0 | 34.8 | 35.7 | 41.0 | **48.2** |

**Discussion**. We previously mentioned that our PDDM and feature embedding can be learned by only optimizing the metric loss $E_m$ in Eq. (3). Here we experimentally prove the necessity of extra supervision from the embedding loss $E_e$ in Eq. (4). Without it, the Recall@K=1 of image retrieval by our 'PDDM score' and 'PDDM+Quadruplet' methods drop by 3.4%+ and 6.5%+, respectively. Another important parameter is the batch size $m$. When we set it to be smaller than 64, say 32, Recall@K=1 on CUB-200-2011 drops to 55.7% and worse with even smaller $m$. This is because the chosen hard quadruplet from a small batch makes little sense for learning. When we use large $m$=132, we have marginal gains but need many more epochs (than 400) to use enough training quadruplets.

**Transfer learning**. Considering the good performance of our fully learned features, here we evaluate their generalization ability under the scenarios of transfer learning and zero-shot learning. Transfer learning aims to transfer knowledge from the source classes to new ones. Existing methods explored the knowledge of part detectors [6] or attribute classifiers [17] across classes. Zero-shot learning is an extreme case of transfer learning, but differs in that for a new class only a description rather than labeled training samples is provided. The description can be in terms of attributes [17], WordNet hierarchy [21, 25], semantic class label graph [10, 24], or text data [8, 22]. These learning scenarios are also related to the open set one [2] where new classes grow continuously.

For transfer learning, we follow [21] to train our feature embeddings and a Nearest Class Mean (NCM) classifier [21] on the large-scale ImageNet 2010[1] dataset, which contains 1,000 classes and more than 1.2 million images. Then we apply the NCM classifier to the larger ImageNet-10K [5] dataset with 10,000 classes, thus do not use any auxiliary knowledge such as parts and attributes. We use the standard flat top-1 accuracy as the classification evaluation metric. Table 2 shows that our features outperform state-of-the-art methods by a large margin, including the deep feature-based ones [18, 21]. We attribute this advantage to our end-to-end feature embedding learning and its large margin nature, which directly translates to good generalization ability.

For zero-shot learning, we follow the standard settings in [21, 25] on ImageNet 2010: we learn our feature embeddings on 800 classes, and test on the remaining 200 classes. For simplicity, we also use the ImageNet hierarchy to estimate the mean of new testing classes from the means of related training classes. The flat top-5 accuracy is used as the classification evaluation metric. As can be seen from Table 2, our features achieve top results again among many competing deep CNN-based methods. Considering our PDDM and local similarity-aware feature embedding are both well learned with safe margins between classes, in this zero-shot task, they would be naturally resistant to class boundary confusion between known and unseen classes.

## 5 Conclusion

In this paper, we developed a method of learning local similarity-aware deep feature embeddings in an end-to-end manner. The PDDM is proposed to adaptively measure the local feature similarity in a heterogeneous space, thus it is valuable for high quality online hard sample mining that can better guide the embedding learning. The double-header hinge loss on both the similarity metric and feature embedding is optimized under the large margin criterion. Experiments show the efficacy of our learned feature embedding in challenging image retrieval tasks, and point to its potential of generalizing to new classes in the large and open set scenarios such as transfer learning and zero-shot learning. In the future, it is interesting to study the generalization performance when using the shared attributes or visual-semantic embedding instead of ImageNet hierarchy for zero-shot learning.

**Acknowledgments**. This work is supported by SenseTime Group Limited and the Hong Kong Innovation and Technology Support Programme.

## Footnotes

[1] See http://www.image-net.org/challenges/LSVRC/2010/index.

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
