[Reviews · NeurIPS 2016]

Reviewer 1

Summary

This paper addresses the problem of learning deep feature embeddings. A key challenge is the non-uniform density of the feature space, which makes effective hard negative mining difficult. The paper proposes a unit that takes into account the mean position of a pair of features (inspired by [35]), along with their difference, in computing a similarity between the feature pair. The positional information allows the unit to account for the non-uniform density in feature space. The unit is differentiable, and can be plugged into existing CNNs. A double-header hinge loss is formulated over pairwise differences/similarities before and after the unit. For each minibatch of SGD, hard quadruplets are mined, where a pair of positive examples with low similarity are found, and corresponding negative pairs with high similarity are found. The approach is evaluated on the task of image retrieval over the CUB-200-2011 and CARS196 datasets, and on transfer/zero-shot learning on ImageNet-10K and ImageNet 2010 datasets with improvements shown for all tasks, particularly with respect to training time.

Qualitative Assessment

The paper addresses a known important problem in computer vision, and the formulation is novel as far as I'm aware. I found the paper to be well written and references good. The results look interesting, and the combination of the PDDM unit and quadruple hard mining appear to offer benefit. I'm a bit confused by the "PDDM score" versus the "Quadruplet+PDDM" rows in Table 1. Just to clarify, do they correspond to retrieval using the output of the PDDM module (former) versus the learned embedding before the PDDM module using Euclidean distance (latter)? I'm curious how important is having the embedding loss E_e in Equation (4) since it appears that gradients can be backpropped through the PDDM modules. Are the features for hard negative mining computed every minibatch round, or are they cached and refreshed from time to time for efficiency?

Confidence in this Review

2-Confident (read it all; understood it all reasonably well)


Reviewer 2

Summary

This paper presents a novel method for local similarity-aware embedding using a deep neural net. The novelty of the paper includes the capability to choose hard samples in training to speed up convergence and boost up performance, use of new objective called double-header hinge loss to learn local structures in the data, use of absolute position information for heterogeneous feature distribution, and use of quadruplet in mini-batch for fast sampling procedure. This paper has shown that the presented method outperforms the other competitors on image retrieval and transfer learning. Major comments: 1. The paper claims that the proposed method has advantages both in performance and speed. However, running time is not reported in the paper. 2. The architecture in FIg. 2 should be explained in more detail. For example, CNN in Fig. 2(a) needs to be defined. 3. More explanation about Fig 3. would be nice. What are the differences among contrastive embedding, triplet embedding, and lifted structured embedding, and when they work and when they do not work in comparison to the presented method? 4. Details of objective optimization is missing. E.g., what is gradient? 5. How does the performance change according to the mini-batch sizes? 6. Are there any issues on initialization of parameters, and how were they initialized in the experiments. 7. Standard deviation of the results in Table 1. would be helpful to see the statistical significance of the results.

Qualitative Assessment

The paper needs more details in the technical parts including optimization, how it was implemented, network architecture to the extent that the results can be reproduced by other researchers. Also, results section should verify each claim made in the paper (e.g., speed).

Confidence in this Review

2-Confident (read it all; understood it all reasonably well)


Reviewer 3

Summary

This paper makes 2 major changes to the deep embedding pipeline. (1) The proposed PDDM leverages the information of absolute feature position to learn a similarity metric adaptive to local feature structure. (2) The proposed double-header hinge loss explicitly utilizes hard quadruplet mining to discriminate the target similarity distributions. Experiments show that the proposed method lead to faster convergence, better feature embedding and generalization ability.

Qualitative Assessment

(1) The paper is well motivated. A global euclidean metric could be suboptimal in a heterogeneous space, which can mislead the hard sample mining and consequently deep embedding learning. Inspired by previous work [Xiong 2012], this paper proposes to leverage the feature position information to learn a locally adaptive similarity metric, which can be used to select genuinely hard samples and guide the deep embedding learning. Although not significantly novel, this sounds reasonable and worth exploring to me. (2) The paper is well presented. The network is well designed to incorporate additional feature position information. The loss function is well designed to explicitly require a hard quadruple mining process. Technical details and hyper-parameters are provided for repeatability. The experiment results are promising compared to previous methods where no feature position information is used.

Confidence in this Review

2-Confident (read it all; understood it all reasonably well)


Reviewer 4

Summary

The authors propose to capture local feature similarity metric using PDDM, to tackle the issues brought by global distance metrics. They also introduce the large-margin double-header hinge loss to jointly optimize the similarity metric learning. They show good performance on image retrieval and transfer learning tasks to demonstrate the efficacy of their method.

Qualitative Assessment

The paper is well written with clear logic and well-designed figures, and shows promising results in both image retrieval and transfer learning tasks. The authors demonstrate clearly the effectiveness of combining distance metric with local similarity achieved by PDDM mechanism. The contribution of PDDM mechanism shows a novel local similarity metric which combines feature difference and absolute position information from [35], and presents surprising fast training convergence and better performance. The contribution of two double-header hinge losses in Eq (4) which combine local similarity metric and distance metric on the hard quadruplet has similar idea with [27], but uses less samples and has low computational cost and faster convergence. Questions for authors: 1) For better understand and self-inclusive of the paper, there should be more formalized description of the comparative methods (i.e. contrastive embedding, triplet embedding and lifted structured embedding) to emphasize the improvement and advantage of the proposed method besides the comparison in Figure 3. 2) The authors should give more analysis or demonstration to explain the key reason of the effectiveness brought by the concatenation of feature difference vector and feature mean vector in the PDDM unit, which could be beneficial and instructive to readers. 3) The authors mention that the whole network, which contains 4 identical CNNs and PDDM units, is trained end-to-end. However, the hard quadruplet mining during a forward pass also needs 2 CNNs and PDDM units. The authors should add more information on the particular forward pass for easy understanding of the end-to-end architecture. 4) I want to confirm that you choose ONLY ONE hard quadruplet ($\hat{i}$, $\hat{j}$, $\hat{k}$, $\hat{l}$) for training per mini-batch (64 samples). If so, you have very few samples to fine-tune the CNN part and train the fully-connected layers of PDDM unit per epoch. I wonder if the samples are enough for training the network parameters on CUB-200-2011 dataset (5,864 training images). Please include more details about this. There are a few typos which need to be amended: 1) Line 47: learn an local-adaptive -> learn a local-adaptive 2) Line 103: into an feature space -> into a feature space 3) Eq (2). $u_1$ and $v_1$ should be $u'$ and $v'$

Confidence in this Review

2-Confident (read it all; understood it all reasonably well)


Reviewer 5

Summary

This paper proposes a framework to learn better deep feature embedding, which is important for the applications like image retrieval and distance-based classification. This framework consists of a newly defined metric called Position-Dependent Deep Metric (PDDM), which considers not only the difference between two features but also their mean, making this metric position-aware. Also, this PDDM helps to identify high quality hard negative samples that are critical for training a metric. Two losses on the PDDM metric and feature embeddings are combined to perform joint metric and embedding optimization. Experimental study on fine-grained image retrieval data set and the tasks of transfer learning and zero-shot learning is performed to demonstrate the efficacy of the proposed framework.

Qualitative Assessment

The proposed deep feature embedding framework is interesting and novel. The idea of location-aware similarity metric is sound. Although learning local distances to better capture the variation of feature space is not new in metric learning, applying this idea to deep feature embedding in this way seems to be novel. The formulation of PDDM is clear and the experimental study demonstrates the advantage of the proposed framework. Comments: 1. Essentially, the requirement of having such a local similarity shall be because the learned deep features are still not sufficiently consistent with the semantic concept (as shown in Figure 1(a)). In this case, will the advantage and improvement of the proposed framework gradually diminish, when a better deep learning network is trained (thus better deep feature representation is obtained)? Please comment. 2. In Eq.(2), u1 and v1 are supposed to be u' and v'? 3. It is better to give an illustration for the description of quadruplet (line 163 to 171) or link it to Figure 1(c). 4. The relationship between S(.,.) and D(.,.) functions could be better explained, which will help to understand the relationship of the two losses of Em and Ee. 5. Experimental study is a bit limited. Comparison on more benchmark data sets (say, Fine-grained image retrieval) and with more existing methods will further enhance this work. After the rebuttal: Thank the authors for the rebuttal. After reading the comments of other reviewers and the rebuttal, I think this work is interesting and novel and the experimental study can demonstrate the effectiveness. I would like to maintain the original rating.

Confidence in this Review

2-Confident (read it all; understood it all reasonably well)


Reviewer 6

Summary

This paper proposes an end-to-end deep feature embedding method by combining similarity metric learning and hard sample mining. The authors use the feature difference vector as well as mean vector to compute the similarity between two feature embeddings. For hard sample mining, they select the hard positive pair with the largest distance from the same class, and then select the sample with the smallest distance from the different classes as the hard negatives for each sample of the selected positive pair. The method achieves promising performance on fine-grained image retrieval, transfer learning and zero-shot learning.

Qualitative Assessment

This paper is well-written and easy to read. The performance of the proposed method is promising on three vision tasks. My main concerns are as follows. 1. They use the feature mean vector to incorporate absolute position information in metric leaning. The idea is already introduced by Xiong et al. [35], which limits the novelty of the proposed PDDM method. 2. They claim they use a better local Euclidean distance metric for hard sample mining by selecting the most dissimilar positive pair and the most similar negative pairs in each batch. It is not efficient for learning since only 4 samples are used in a batch (e.g., with batch size of 64). More importantly, they do not evaluate the impact of the important parameter of the batch size in this method. 3. There are no discussions on how to choose the other algorithmic parameters. For example, no experimental analysis of lambda makes the effectiveness of the feature-level metric learning unclear. 4. Why will selecting the hardest samples not lead to bad local minima in this method since hard sample mining is performed throughout the learning process not just in early training iterations? It would be interesting to show the performance without hard sample mining in this method. 5. The method treats all the visual classes equal and unrelated without considering the semantic relationship among them. For example, cat and dog should be closer than cat and bicycle in the semantic space. I wonder whether the semantic information helps learn better feature embeddings in this method. 6. Typos: u_1 and v_1 should be u' and v' in Eq. 2.

Confidence in this Review

3-Expert (read the paper in detail, know the area, quite certain of my opinion)